

# Occult zonulopathy detected during cataract surgery in patients with acute primary angle closure: a retrospective study

Jiawei Chen[1,2,3], Xiang-Ling Yuan[1,4], Xinyue Zhang[1,2], Yanjun Huang[3], Xiaona Huang[3] and Xuanchu Duan[1,2]

[1] Aier Academy of Ophthalmology, Central South University, Changsha, Hunan, China
[2] Aier Glaucoma Institute, Hunan Engineering Research Center for Glaucoma with Artificial Intelligence in Diagnosis and Application of New Materials, Changsha Aier Eye Hospital, Changsha, Hunan, China
[3] Department of Ophthalmology, The Sixth Affiliated Hospital of South China University of Technology, Foshan, Guangdong, China
[4] Aier Eye Institute, Changsha, Hunan, China

## ABSTRACT

**Background**. Whether occult zonulopathy contributes to the development of acute primary angle closure (APAC) remains elusive. This study aimed to determine the association between occult zonulopathy detected during cataract surgery and APAC and to investigate the biometric characteristics of APAC patients with or without occult zonulopathy.

**Methods**. Retrospective case-control study. A total of 27 Chinese unilateral APAC subjects and 132 control subjects with comprehensive ophthalmic examinations were recruited. Occult zonulopathy was identified with the intraoperative signs during cataract surgery. The proportion of occult zonulopathy was compared between the APAC and control groups. A multivariate logistic analysis was conducted to determine the association between occult zonulopathy and APAC. The ocular biometric parameters were compared between APAC and the contralateral eyes in APAC patients with or without occult zonulopathy.

**Results**. APAC patients (63.0%) had a significantly larger proportion of occult zonulopathy than control subjects (1.5%, $P < 0.001$). In the multivariate logistic analysis, occult zonulopathy was significantly associated with APAC after adjusting the axial length (AL) and sex (OR = 126.49, 95% CI [20.89–766.02]; $P < 0.001$). Compared to contralateral eyes, shallower central anterior chamber depth, more anterior lens position and relative lens position were found in APAC eyes both with and without occult zonulopathy (all $P < 0.05$), but no difference in AL and lens thickness.

**Conclusion**. A larger proportion of occult zonulopathy was significantly associated with APAC. Occult zonulopathy could be a risk factor for APAC by inducing forward shifting of the lens.

Corresponding author
Xuanchu Duan,
duanxchu@csu.edu.cn

## INTRODUCTION

Glaucoma, a leading cause of irreversible blindness, brings about an increasing burden worldwide (*GBD 2019 Blindness and Vision Impairment Collaborators, 2021*; *Lin et al., 2023*). Primary angle-closure glaucoma (PACG) is one of its essential subtypes, characterized by an anatomically closed angle resulting in intraocular pressure (IOP) elevation (*Weinreb, Aung & Medeiros, 2014*). PACG is responsible for approximately 50% of blindness attributed to glaucoma (*Quigley & Broman, 2006*). It was estimated that PACG would affect more than 32 million individuals globally by 2040, with about 80% of these cases in Asia (*Tham et al., 2014*). According to the protocol of the International Society of Geographic and Epidemiologic Ophthalmology (ISGEO), primary angle closure was categorized into three types: primary angle-closure suspect (PACS), primary angle closure (PAC), and primary angle-closure glaucoma (PACG) (*Foster et al., 2002*). Acute primary angle closure (APAC) is considered an ophthalmic emergency with a rapid increase in IOP, which could induce acute optic nerve injury and potentially lead to permanent visual loss or blindness (*Gedde et al., 2021*).

Studies have shown that short axial length (AL), shallow anterior chamber depth (ACD), and thick lens thickness (LT) are the collective anatomical risk factors for PACG (*Marchini et al., 2015*). Pupillary block, plateau iris, abnormal lens position, and choroidal effusion are taken for the main mechanisms of closed angle in PACG (*Sun et al., 2017*). Among patients diagnosed with unilateral APAC, only 6.5% developed glaucoma in their contralateral eyes during a follow-up period of 4 to 10 years (*Friedman et al., 2006*). It remains confusing that some unilateral APAC patients could avoid acute angle-closure attacks on their contralateral eyes for years without any treatment. Rather than acute angle closure, most angle closure developed chronically wanting in signs and symptoms, which is still difficult to fully interpret by the available evidence (*Wilensky et al., 1993*). With the comprehensive application of phacoemulsification in cataract extraction, a significant incidence of occult zonulopathy in PACG patients was observed during cataract surgery (*Kwon & Sung, 2017*; *Salimi et al., 2021*; *Zhang et al., 2023*). However, limited studies were conducted to demonstrate the association between occult zonulopathy and APAC. Additionally, previous studies have shown inconsistent findings on the association between a relatively longer axial length and zonulopathy (*Kwon & Sung, 2017*; *Zhang et al., 2023*). For the critical impact of lens zonules and lens position on the development of PACG, clarifying the relationship between occult zonulopathy and acute angle closure would help elucidate the unpredictable occurrence of APAC, understand the pathophysiology of asymmetric APAC, and guide the personalized treatment. Hence, this study aimed to investigate the potential links between occult zonulopathy and APAC and to explore the biometric characteristics of APAC patients with or without occult zonulopathy. Due to the limitation of the retrospective study design, the causality between occult zonulopathy and APAC was not addressed in this study.

## MATERIALS & METHODS

### Participants

This was a retrospective, case-control study following the tenets of the Declaration of Helsinki. Approval was granted by the Medical Ethics Committee of the Sixth Affiliated Hospital of South China University of Technology (Foshan Nanhai District People's Hospital; approval number: 2023008). After an explanation of the nature and possible consequences of the study, written informed consent was obtained from all subjects. This study was registered in the Chinese Clinical Trial Registry (registration number: ChiCTR2300077395). According to the previous studies, the proportion of the intraoperative zonulopathy were 69% in APAC patients (*Zhang et al., 2023*) and 10.9% in patients with age-related cataract (*Zhang et al., 2024*) respectively. With $\alpha = 0.05$, $\beta = 0.01$, and ratio = 4, sample sizes of no less than 13 in APAC group and 52 in Control group were needed to detect a difference between the group proportions.

A total of 27 unilateral APAC subjects and 132 control subjects were recruited from April 1, 2022, to June 1, 2023 (Fig. 1). APAC was diagnosed according to the following criteria (*Aung et al., 2004*; *Li et al., 2023*): (1) at least two of the following symptoms: ophthalmalgia or periocular pain, nausea and/or vomiting, an antecedent history of intermittent blurring of vision with halos; (2) acute increase in IOP (>30 mmHg); (3) presenting at least one of the three signs: conjunctival injection, corneal epithelial edema, glaucomflecken, and mild-dilated unreactive pupil; (4) presenting shallow anterior chamber in both eyes, with a closed angle in APAC eye and a narrow-angle in the fellow eye under gonioscopy. APAC patients who were accompanied by age-related cataract with best-corrected visual acuity (BCVA) worse than 0.3 logarithms of the minimum angle of resolution (logMAR) were included in this study. The control subjects were enrolled as follows: (1) age above 50 years; (2) diagnosed as age-related cataract with BCVA worse than 0.3 logMAR; (3) open-angle in gonioscopy with IOP $\leq$ 21 mmHg without medications; (4) absent of glaucomatous optic neuropathy or visual field damage.

To diminish potential confounding effect, patients with one of the following situations were excluded: (1) history or signs of acute angle closure in the contralateral eyes of APAC eyes; (2) history of surgical or laser peripheral iridectomy, laser peripheral iridoplasty, trabeculectomy, vitrectomy, pterygium excision, and so on; (3) history or signs of ocular trauma and traumatic surgery in the eye; (4) history of high myopia (axial length $\geq$26 mm), chronic PACG, primary open-angle glaucoma, secondary glaucoma, pseudoexfoliation syndrome, uveitis, retinal detachment, retinitis pigmentosa, diabetic retinopathy, hypermature cataract, ocular tumor, Marfan syndrome, Marchesani syndrome, *etc.*; (5) diagnosis or signs of lens subluxation or luxation (iridodonesis, phacodonesis, visibility of lens equator). Occult zonulopathy was defined by the sign NO. 1 combined with the sign NO. 2, or the sign NO. 1 combined with the sign NO. 3 among the following intraoperative signs during cataract surgery (*Qiao et al., 2022*; *Zhang et al., 2023*): (1) wrinkling of anterior lens capsules while making continuous curvilinear capsulorhexis; (2) distorted anterior lens capsule opening or a floppy capsular bag after cortical removal; (3) visualization of the capsular equator during or after nuclear/cortical removal (Fig. 2).

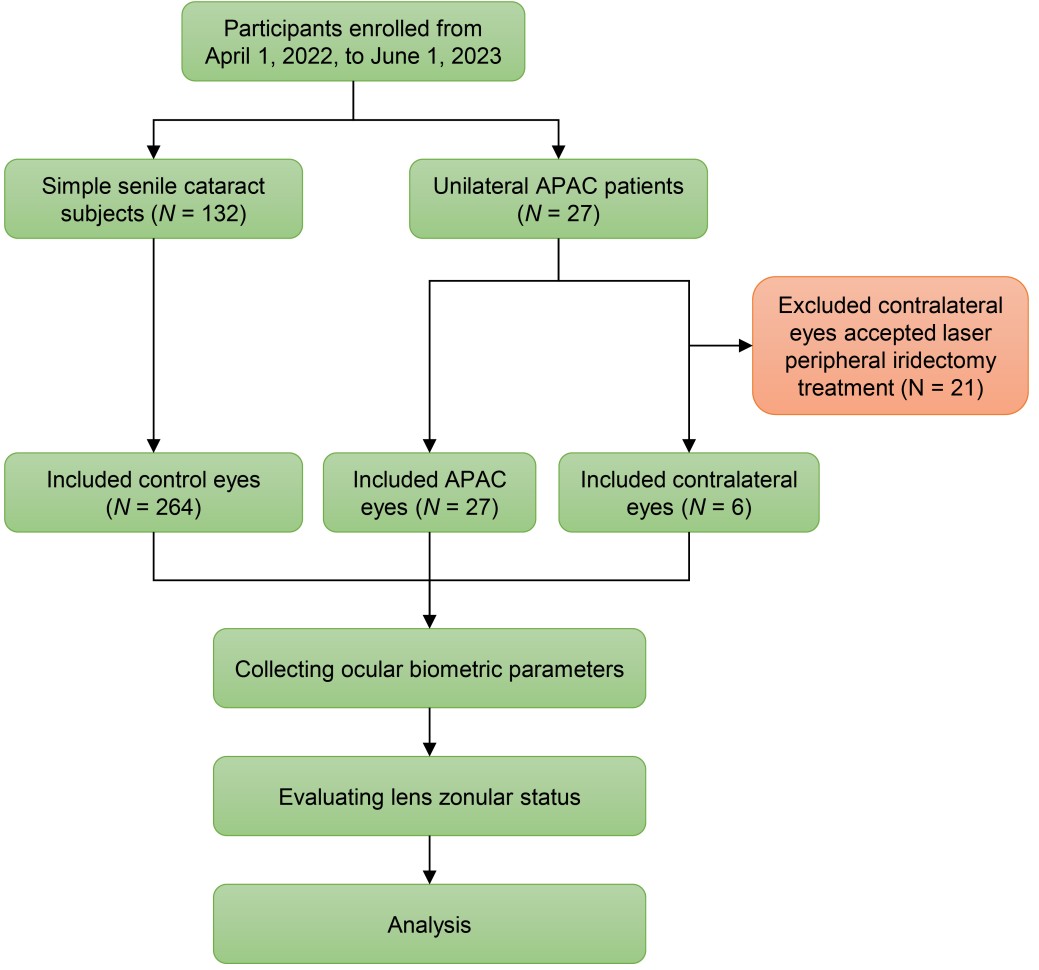

**Figure 1** **Flow diagram showing study design and subject inclusion.**

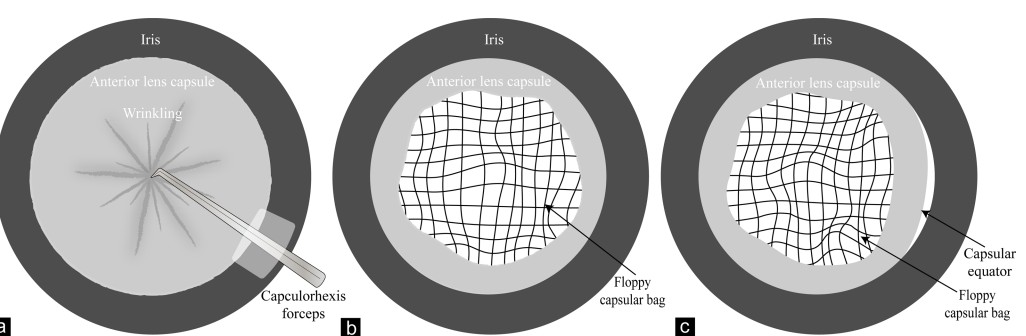

**Figure 2** **Diagrams of intraoperative signs of occult zonulopathy.** (A) Wrinkling of anterior lens capsules while making continuous curvilinear capsulorhexis; (B) a distorted anterior lens capsules opening and floppy capsular bag after cortical removal; (C) a distorted anterior lens capsules opening, floppy capsular bag and visualization of the capsular equator after cortical removal.

## Ophthalmic examinations

All participants underwent detailed ophthalmologic examinations before surgery, including lamp biomicroscope examination, best-corrected visual acuity measurement, IOP measurement, optical coherence tomography measurement (Zeiss Cirrus HD-OCT 500, Carl Zeiss; Jena, Germany), visual field test (Humphrey Field analyzer II, Carl Zeiss Humphrey 750i; Jena, Germany), and ocular biometric measurement (IOL master 700, Carl Zeiss; Jena, Germany). Anti-glaucoma medications (brimonidine, brinzolamide, timolol, mannitol, pilocarpine) and prednisolone acetate eye drops were prescribed to APAC patients for lowering IOP and anti-inflammation. All the ophthalmological examinations were performed under a transparent cornea and IOP $\leq 30$ mmHg.

## Data collection

Data recorded from the medical record system including age, sex, medical histories, diagnosis, and surgical records of all subjects were obtained. Ocular biometric data of AL, flat keratometry, steep keratometry, central ACD, LT (lens thickness), central corneal thickness (CCT), corneal diameter, pupil diameter, angle alpha, degree of alpha, angle kappa, and degree of kappa were collected from the ocular biometric measurements. The ACD did not contain CCT. The lens position (LP = ACD + 1/2 LT) and relative lens position (RLP = (ACD +1/2 LT)/AL) were calculated.

## Statistical analyses

Only the primary surgical eye of each bilateral control subject was included in the comparison of biometric parameters with APAC eyes. Shapiro–Wilk test was used to assess data distribution normality. Continuous variables were described as mean with standard deviation (mean $\pm$ SD) or median with interquartile range (IQR: Q1, Q3). An independent $t$-test was used to compare normality-distributed continuous data between groups, and the Mann–Whitney $U$ test was applied to compare disnormality-distributed continuous data. The paired $t$-test or Wilcoxon-matched rank test was performed to compare the biometric parameters between the APAC and contralateral eyes. Categorical variables were presented as frequency with percentage and compared by $\chi^2$ test or Fisher's exact test. Univariate and multivariate logistic analyses were performed to explore the association of biometric parameters and occult zonulopathy with APAC. The difference was considered statistically significant with $P < 0.05$ or $P < 0.05/3 = 0.0167$ in multiple comparisons after Bonferroni's correction. All statistical analyses were performed using SPSS STATISTICS 26.0 (IBM SPSS Inc., Chicago, IL, USA). Sample size and statistical power was calculated using PASS 15.0.5 (http://www.ncss.com; NCSS, LLC. Kaysville, Utah, USA).

## RESULTS

Twenty-seven APAC eyes with 27 contralateral eyes and 264 eyes of 132 control subjects were included in the present study. Twenty-seven APAC eyes, six contralateral eyes, and 264 eyes of control subjects underwent phacoemulsification cataract extraction and intraocular lens implantation (or combined with goniosynechialysis in APAC) conducted by one

of three specialist surgeons. There were no significant differences in occult zonulopathy among surgeons ($P = 0.100$). No significant difference in age was found between the APAC group ($67.26 \pm 9.24$ years) and the control group ($70.58 \pm 7.88$ years, $P = 0.054$) (Table 1). The APAC group (85.2%) had significantly more females than the control group (61.4%, $P = 0.025$). The APAC eyes showed significantly shorter AL ($22.54 \pm 0.73$ mm, $P < 0.001$), shallower central ACD ($1.57 \pm 0.26$ mm, $P < 0.001$), thicker LT ($5.09 \pm 0.37$ mm, $P < 0.001$), more anterior LP ($4.12 \pm 0.21$ mm, $P < 0.001$) and RLP ($18.29 \pm 0.91\%$, $P < 0.001$), thicker CCT ($571.33 \pm 62.11$ $\mu$m, $P = 0.002$), and smaller corneal diameter ($11.24 \pm 0.46$ mm, $P = 0.001$) as compared to control eyes. Contralateral eyes of APAC eyes had significantly shorter AL, shallower central ACD, thicker LT, more anterior LP and RLP as compared to the control group (all $P < 0.001$). Compared with contralateral eyes (central ACD: $1.72 \pm 0.27$ mm; LP: $4.26 \pm 0.21$ mm; RLP: $18.98 \pm 0.99\%$), APAC eyes showed significantly shallower central ACD ($1.57 \pm 0.26$ mm, $P = 0.008$), more anterior LP ($4.12 \pm 0.21$ mm, $P = 0.007$) and RLP ($18.29 \pm 0.91\%$, $P = 0.006$), but no significant difference in AL and LT. APAC eyes ($0.62 \pm 0.46$ mm) had a significantly greater angle kappa than control ($0.34 \pm 0.27$ mm, $P < 0.001$) and contralateral eyes ($0.36 \pm 0.30$ mm, $P = 0.011$) (Fig. S1).

The percentage of occult zonulopathy was significantly higher in APAC eyes (63.0%) than in control eyes (1.5%, $P < 0.001$) and the contralateral eyes ($P = 0.007$). There were no significant differences in occult zonulopathy between the contralateral eyes and control eyes ($P = 1.00$). In the multivariate logistic analysis, occult zonulopathy was significantly associated with APAC after adjusting sex and AL (odds ratio (OR) = 126.49, 95% CI [20.89–766.02]; $P < 0.001$) (Table 2). We achieved 100% statistical power to detect the significant findings using a Chi-Square test with a 0.05 significance level ($\alpha = 0.05$, $P_{(probability\ of\ exposure\ among\ controls)} = 0.015$, $N_{(cases)} = 27$, and $M_{(controls\ per\ case)} = 4$). The ACD, LT, LP and RLP were excluded from multivariate logistic analysis because of their potential correlation with AL. Compared to contralateral eyes, shallower central ACD, thicker CCT, more anterior LP and RLP were observed in both APAC eyes with occult zonulopathy and without occult zonulopathy (all $P < 0.05$) (Table 3) For APAC patients, eyes with occult zonulopathy showed no significant difference in biometric parameters compared to eyes without occult zonulopathy (all $P > 0.05$) (Fig. S2).

For the binocular differences in ocular biometric parameters, APAC patients showed significantly larger variations in steep keratometry ($P < 0.001$), LP ($P < 0.001$), RLP ($P < 0.001$), CCT ($P < 0.001$), corneal diameter ($P = 0.005$), pupil diameter ($P < 0.001$), and angle kappa ($P < 0.001$) than control subjects, but the more minor binocular difference in LT ($P = 0.016$) (Table 4). The binocular difference in LT of APAC subjects with occult zonulopathy (median: 0.070, IQR: 0.035 to 0.150) was significantly greater than that of APAC eyes without occult zonulopathy (median: 0.025, IQR: 0.018 to 0.073; $P = 0.046$). No binocular differences were detected in AL, keratometry, central ACD, LP, RLP, CCT, corneal diameter, pupil diameter, angle alpha, and angle kappa between APAC eyes with and without occult zonulopathy (all $P > 0.05$).

**Table 1  Demographic and biometric parameters of control, APAC eyes and the contralateral eyes.**

| | Control eyes G1 ($n = 132$) | APAC eyes G2 ($n = 27$) | Contralateral eyes G3 ($n = 27$) | P G1 vs G2 | P G1 vs G3 | P G2 vs G3 |
|---|---|---|---|---|---|---|
| Age (years) | 70.58 ± 7.88 | 67.26 ± 9.24 | – | 0.054[‡] | – | – |
| Sex | | | | | | |
|    Male (%) | 51 (38.6%) | 4 (14.86%) | – | 0.025[*] | – | – |
|    Female (%) | 81 (61.4%) | 23 (85.2%) | | | | |
| Occult zonulopathy | | | | | | |
|    No (%) | 130 (98.5%) | 10 (37.0%) | 6 | <0.001[†] | 1.000[†] | 0.007[†] |
|    Yes (%) | 2 (1.5%) | 17 (63.0%) | 0 | | | |
| Axial length (mm) | 23.49 ± 0.98 | 22.54 ± 0.73 | 22.45 ± 0.74 | <0.001[‡] | <0.001[‡] | 0.561[§] |
| Flat keratometry (diopters) | 43.79 ± 1.63 | 43.75 ± 1.34 | 43.88 ± 1.28 | 0.912[‡] | 0.795[‡] | 0.539[§] |
| Steep keratometry (diopters) | 44.91 ± 1.78 | 44.97 ± 1.40 | 44.80 ± 1.32 | 0.869[‡] | 0.761[‡] | 0.340[§] |
| Central ACD (mm) | 2.51 ± 0.40 | 1.57 ± 0.26 | 1.72 ± 0.27 | <0.001[‡] | <0.001[‡] | 0.008[§] |
| Lens thickness (mm) | 4.58 ± 0.48 | 5.09 ± 0.37 | 5.07 ± 0.35 | <0.001[‡] | <0.001[‡] | 0.364[§] |
| Lens position (mm) | 4.80 ± 0.30 | 4.12 ± 0.21 | 4.26 ± 0.21 | <0.001[‡] | <0.001[‡] | 0.007[§] |
| Relative lens position (%) | 20.43 ± 1.07 | 18.29 ± 0.91 | 18.98 ± 0.99 | <0.001[‡] | <0.001[‡] | 0.006[§] |
| Central corneal thickness (μm) | 528.27 ± 32.67 | 571.33 ± 62.11 | 528.26 ± 34.36 | 0.002[‡] | 0.999[‡] | <0.001[§] |
| Corneal diameter (mm) | 11.64 ± 0.52 | 11.24 ± 0.46 | 11.40 ± 0.34 | 0.001[‡] | 0.038[‡] | 0.050[§] |
| Pupil diameter (mm) | 3.59 ± 0.87 | 3.82 ± 1.49 | 2.70 ± 1.06 | 0.441[‡] | <0.001[‡] | 0.001[§] |
| Angle alpha (mm) | 0.57 ± 0.32 | 0.73 ± 0.37 | 0.58 ± 0.30 | 0.012[‡‡] | 0.969[‡‡] | 0.074[‖] |
| Angle kappa (mm) | 0.34 ± 0.27 | 0.62 ± 0.46 | 0.36 ± 0.30 | <0.001[‡‡] | 0.713[‡‡] | 0.011[‖] |

**Notes.**

APAC, acute primary angle closure; ACD, anterior chamber depth; n, number.

Lens position, ACD + 1/2 lens thickness; Relative lens position, lens position/axial length.

Statistically significant difference defined as $P < 0.05/3 = 0.0167$.

[*] $\chi 2$ test.

[†] Fisher's exact test.

[‡] Independent $t$ test.

[‡‡] Mann-Whitney $U$ test.

[§] Paired $t$ test.

[‖] Wilcoxon-matched rank test.

# DISCUSSION

Lens zonules are critical anatomic structures for lens stabilization. Severe zonulopathy could result in lens luxation or subluxation with typical signs, such as iridodonesis, phacodonesis, asymmetric ACD in both eyes, and even lens subluxation (*Jing et al., 2021*; *Zhang et al., 2019*). Lens luxation or subluxation is the common cause of secondary angle-closure glaucoma (*Chen et al., 2023b*; *Tang et al., 2024*; *Xing et al., 2020*). Unlike lens luxation or subluxation, occult zonulopathy is a result of weakness or partial damage in lens zonules without evident preoperative signs of lens luxation or subluxation. Our study showed that only a very small proportion of normal cataract patients would be affected by occult zonulopathy. *Zhang et al. (2024)* found that 10.9% of patients with age-related cataract were diagnosed with zonulopathy intraoperatively, which was obviously higher than our finding on occult zonulopathy (1.5%) because they included patients with visualization of the equator of the lens with fully dilated pupil and out of shape or deviation of the anterior capsular opening after cortex removed.

**Table 2  Univariate and multivariate logistic analysis of occult zonulopathy with APAC.**

|  | Univariate logistic analysis | | Multivariate logistic analysis | |
| --- | --- | --- | --- | --- |
|  | OR (95% CI) | *P* | OR (95% CI) | *P* |
| Age (per year) | 0.95 (0.90–1.00) | 0.057 | – | |
| Sex (female) | 3.62 (1.18–11.08) | 0.024 | 2.55 (0.38–16.90) | 0.333 |
| Occult zonulopathy (yes) | 110.50 (22.31–547.38) | <0.001 | 126.49 (20.89–766.02) | <0.001 |
| Axial length (mm) | 0.32 (0.18–0.56) | <0.001 | 0.32 (0.14–0.72) | 0.006 |
| Flat keratometry (diopters) | 0.99 (0.76–1.28) | 0.912 | – | |
| Steep keratometry (diopters) | 1.02 (0.80–1.29) | 0.868 | – | |
| Central ACD (per 0.1mm) | 0.43 (0.30–0.62) | <0.001 | – | |
| Lens thickness (mm) | 11.81 (3.92–35.59) | <0.001 | – | |
| Lens position (per 0.1mm) | 0.19 (0.09–0.41) | <0.001 | – | |
| Relative lens position (%) | 0.04 (0.01–0.14) | <0.001 | – | |
| Central corneal thickness ($\mu$m) | 1.02 (1.01–1.04) | <0.001 | – | |
| Corneal diameter (mm) | 0.27 (0.12–0.64) | 0.003 | – | |
| Pupil diameter (mm) | 1.24 (0.84–1.83) | 0.278 | – | |
| Angle alpha (mm) | 3.38 (1.14–10.08) | 0.029 | – | |
| Angle kappa (mm) | 9.12 (2.63–31.60) | <0.001 | – | |

**Notes.**

APAC, acute primary angle closure; ACD, anterior chamber depth; n, number.

Lens position, ACD + 1/2 lens thickness.

Relative lens position, lens position/axial length.

To determine the unclear association between occult zonulopathy and APAC, we conducted this respective study and revealed a significantly higher incidence of occult zonulopathy in APAC eyes (63.0%) than the normal eyes with age-related cataract, which was close to the reported incidence in a previous study (69%) (*Zhang et al., 2023*). Twenty-one percent of APAC patients had zonular instability intraoperatively in a Korean study (*Kwon & Sung, 2017*). This study may have reported a lower incidence of occult zonulopathy due to excluding APAC patients with intraoperative zonular damage. In a large cohort study, zonulopathy was determined among 7.3% of the 806 eyes with primary angle closure disease (PACD) (*Salimi et al., 2021*). It suggests that the prevalent of occult zonulopathy in APAC is significantly higher than that in PACD. Furthermore, the proportion of occult zonulopathy in APAC eyes was remarkably higher than in the control and contralateral eyes (Table 1), which was consistent with the prior study (*Zhang et al., 2023*). Our study showed that people with occult zonulopathy had over 126 times higher risk of APAC than those without occult zonulopathy despite having the same AL, but it has a relatively high confidence interval range. Overall, the findings of our study suggest that occult zonulopathy could be a crucial risk factor for APAC.

This study revealed that APAC eyes with occult zonulopathy had relatively longer AL, smaller keratometry, shallower central ACD, thinner LT, more anterior LP and RLP than that of APAC eyes without occult zonulopathy, though the differences were not statistically significant (Table 3). Since only occult zonulopathy but not potential lens subluxation or luxation was considered in this study, slight zonular instability may cause only subtle changes in ocular structures. Another study found that shallower ACD and thicker LT,

Chen et al. (2025), *PeerJ*, DOI 10.7717/peerj.19330

**Table 3  Comparison of biometric parameters between APAC eyes with or without occult zonulopathy and the contralateral eyes.**

| | APAC with occult zonulopathy (n = 17) | | APAC without occult zonulopathy (n = 10) | | $P^{\parallel}$ G1 vs G2 | $P^{\parallel}$ G3 vs G4 | $P^{\dagger}$ G1 vs G3 |
|---|---|---|---|---|---|---|---|
| | APAC eyes G1 | Contralateral eyes G2 | APAC eyes G3 | Contralateral eyes G4 | | | |
| Axial length (mm) | 22.64 ± 0.80 | 22.71 ± 0.82 | 22.37 ± 0.58 | 22.32 ± 0.51 | 0.421 | 0.172 | 0.175 |
| Flat keratometry (diopters) | 43.43 ± 1.36 | 43.59 ± 1.33 | 44.30 ± 1.19 | 44.37 ± 1.09 | 1.000 | 0.386 | 0.050 |
| Steep keratometry (diopters) | 44.75 ± 1.28 | 44.63 ± 1.37 | 45.35 ± 1.57 | 45.09 ± 1.25 | 0.523 | 0.959 | 0.248 |
| Central ACD (mm) | 1.56 ± 0.29 | 1.70 ± 0.32 | 1.59 ± 0.22 | 1.75 ± 0.14 | 0.037 | 0.007 | 0.880 |
| Lens thickness (mm) | 5.08 ± 0.44 | 5.05 ± 0.42 | 5.11 ± 0.24 | 5.11 ± 0.21 | 0.618 | 0.683 | 0.782 |
| Lens position (mm) | 4.11 ± 0.23 | 4.23 ± 0.25 | 4.14 ± 0.18 | 4.31 ± 0.12 | 0.019 | 0.005 | 0.802 |
| Relative lens position (%) | 18.14 ± 0.95 | 18.79 ± 1.15 | 18.54 ± 0.84 | 19.31 ± 0.51 | 0.019 | 0.005 | 0.192 |
| Central corneal thickness (μm) | 581.18 ± 72.90 | 530.53 ± 41 | 554.60 ± 34.69 | 524.40 ± 19.87 | 0.001 | 0.019 | 0.393 |
| Corneal diameter (mm) | 11.29 ± 0.49 | 11.42 ± 0.33 | 11.16 ± 0.41 | 11.36 ± 0.37 | 0.161 | 0.096 | 0.449 |
| Pupil diameter (mm) | 3.95 ± 1.61 | 2.53 ± 0.76 | 3.60 ± 1.30 | 2.98 ± 1.45 | 0.004 | 0.343 | 0.421 |
| Angle alpha (mm) | 0.67 ± 0.29 | 0.63 ± 0.34 | 0.84 ± 0.49 | 0.49 ± 0.22 | 0.435 | 0.093 | 0.615 |
| Angle kappa (mm) | 0.52 ± 0.28 | 0.41 ± 0.36 | 0.79 ± 0.65 | 0.27 ± 0.14 | 0.231 | 0.012 | 0.461 |

**Notes.**

APAC, acute primary angle closure; ACD, anterior chamber depth; n, number.

Lens position, ACD + 1/2 lens thickness.

Relative lens position, lens position/axial length.

[†] Mann–Whitney $U$ test.

[‖] Wilcoxon-matched rank test.

Chen et al. (2025), *PeerJ*, DOI 10.7717/peerj.19330

**Table 4 Comparison of the binocular differences in biometric parameters between control and APAC eyes, APAC eyes with and without occult zonulopathy.**

| | Control (*n* = 132) | APAC (*n* = 27) | *P*[†] | APAC with occult zonulopathy (*n* = 17) | APAC without occult zonulopathy (*n* = 10) | *P*[†] |
|---|---|---|---|---|---|---|
| Axial length (mm) | 0.100 (0.043, 0.210) | 0.120 (0.080, 0.170) | 0.377 | 0.120 (0.080, 0.225) | 0.115 (0.078, 0.163) | 0.473 |
| Flat keratometry (diopters) | 0.370 (0.173, 0.588) | 0.480 (0.270, 1.060) | 0.072 | 0.450 (0.290, 0.960) | 0.585 (0.135, 1.143) | 0.941 |
| Steep keratometry (diopters) | 0.260 (0.120, 0.500) | 0.610 (0.230, 0.930) | <0.001 | 0.600 (0.200, 0.990) | 0.625 (0.225, 1.065) | 0.941 |
| Central ACD (mm) | 0.076 (0.037, 0.166) | 0.126 (0.058, 0.279) | 0.060 | 0.126 (0.053, 0.307) | 0.119 (0.048, 0.240) | 0.749 |
| Lens thickness (mm) | 0.095 (0.040, 0.220) | 0.060 (0.020, 0.140) | 0.016 | 0.070 (0.035, 0.150) | 0.025 (0.018, 0.073) | 0.046 |
| Lens position (mm) | 0.059 (0.030, 0.105) | 0.101 (0.061, 0.250) | <0.001 | 0.148 (0.061, 0.250) | 0.100 (0.066, 0.240) | 0.749 |
| Relative lens position (%) | 0.228 (0.114, 0.442) | 0.513 (0.246, 0.107) | <0.001 | 0.577(0.220, 1.884) | 0.484 (0.359, 1.188) | 0.824 |
| Central corneal thickness (μm) | 12.0 (5.0, 18.0) | 35.0 (20.0, 61.0) | <0.001 | 27.0 (15.5, 73.0) | 35.5 (28.3, 51.3) | 0.786 |
| Corneal diameter (mm) | 0.20 (0.10, 0.30) | 0.30 (0.20, 0.50) | 0.005 | 0.30 (0.20, 0.45) | 0.30 (0.10, 0.53) | 0.863 |
| Pupil diameter (mm) | 0.30 (0.10, 0.50) | 1.50 (0.59, 2.40) | <0.001 | 1.90 (0.64, 2.75) | 1.25 (0.35, 2.00) | 0.264 |
| Angle alpha (mm) | 0.14 (0.053, 0.30) | 0.24 (0.05, 0.30) | 0.404 | 0.18 (0.04, 0.29) | 0.28 (0.10, 0.72) | 0.223 |
| Angle kappa (mm) | 0.10 (0.00, 0.20) | 0.20 (0.10, 0.70) | <0.001 | 0.20 (0.15, 0.45) | 0.30 (0.08, 0.80) | 0.711 |

**Notes.**

APAC, acute primary angle closure; ACD, anterior chamber depth; n, number.

Lens position, ACD + 1/2 lens thickness.

Relative lens position, lens position/axial length.

Data were described as median with interquartile range (Q1, Q3).

[†] Mann–Whitney *U* test.

but not AL and RLP, were uncovered in APAC patients with zonulopathy (*Zhang et al., 2023*). *Kwon & Sung (2017)* reported longer AL and higher lens vault (LV) in APAC eyes with zonular instability but no difference in ACD, anterior chamber width (ACW), angle opening distance (AOD) 750, and trabecular iris space area (TISA) 750. In the study by *Chen et al. (2023a)*, APAC eyes with zonular laxity had significantly shallower ACD, higher LV, less RLP, thinner iris thickness (IT), smaller anterior chamber area (ACA) and AOD500, and no significant differences in AL, LT, ACW, anterior placement of the ciliary body (APCB), and ciliary body thickness (CBT) were found. The results widely varied among studies due to the differences in inclusion criteria, sample sizes, and included populations. Regardless, those inconsistent results seem to imply an anteriorly placed thicker lens, shallower anterior chamber, and longer AL in APAC eyes with occult zonulopathy. Further multicentric research with a larger sample size and unitive inclusion criteria is warranted to confirm the discrepancy in ocular biometric parameters between APAC eyes with and without occult zonulopathy.

Several investigations have demonstrated shallower central ACD, LP and RLP closer to the anterior in APAC eyes compared to the contralateral eyes, which was also found in this study (*Li et al., 2018*; *Senthilkumar et al., 2022*; *Zhang et al., 2010*). The smaller values of ACA, AOD, CBT, APCB, TISA, iris area, and iris curvature were found in APAC eyes (*Li et al., 2018*; *Senthilkumar et al., 2022*). Our study showed no significant difference in AL and LT between the bilateral eyes in APAC patients, aligning with previous findings (*Li et al., 2018*; *Moghimi et al., 2014*). Therefore, it seems that a forward movement of the lens could be a reasonable explanation for the crowded anterior structures in APAC eyes under the circumstance of having a similar dimension of eyeball with contralateral eyes. In the present study, APAC eyes were found to have greater angle alpha and kappa than normal eyes and a greater angle kappa than fellow eyes (Table 1). Both greater angle alpha and angle kappa have been demonstrated to be associated with crystalline lens decentration and tilt (*Li et al., 2021*; *Shen et al., 2023*). As the crystalline lens decentration and tilt were related to PAC diseases, greater angle alpha and angle kappa may suggest a change in lens stability in APAC eyes (*Wang et al., 2020*). Occult zonulopathy could fully explain the forward shifting of the lens and the altered lens stability in APAC.

Similar tendency of shallower central anterior chamber depth, more anterior lens position and relative lens position were found in APAC eyes both with and without occult zonulopathy compared to contralateral eyes, suggesting that biometric parameters changes between APAC eye and contralateral eye might not uncover the potential occult zonulopathy in APAC patients. A slight change in biometric parameters attributed to occult zonulopathy may be challenging to detect for the existing crowded anterior segment in APAC patients. APAC with occult zonulopathy had significantly greater binocular differences in LT than APAC without occult zonulopathy, implying that the binocular differences in biometric parameters might be more valuable in predicting occult zonulopathy. APAC patients with occult zonulopathy showed similar binocular differences in AL, central ACD, LP, RLP, angle alpha, and angle kappa compared to those without occult zonulopathy, whereas LP and RLP showed no significant difference in APAC eyes without zonular laxity, and greater binocular differences in LT, LV, RLP, ACD, ACA, AOD500, and

IT were discovered in APAC patients with zonular laxity in a study (*Chen et al., 2023a*). The different inclusion criteria and sample sizes may account for the inconsistent results. APAC patients demonstrated larger binocular differences in steep keratometry, LP, RLP, corneal diameter, and angle kappa of bilateral eyes than control subjects, suggesting the more asymmetric ocular anterior structures in APAC patients. The binocular differences in APAC patients are too small to recognize under regular examinations before surgery.

Although shallow ACD and higher LV were shown as risk factors for zonular laxity, they could result from the forward shifting of the lens (*Chen et al., 2023a*). A longer AL ($23.23 \pm 0.55$ mm) is a most likely risk factor for occult zonulopathy in APAC patients, as it was independent of the abnormal lens position (*Kwon & Sung, 2017*). In addition, the potential role of genetics in the development of occult zonulopathy also deserves more attention. Occult zonulopathy can promote the development of APAC by causing the anterior shifting of the lens, resulting in a more crowded ocular anterior segment. The iris closely contacts the anterior surface of the lens, making pupillary block prone to happen easily (Fig. 3). Rapid IOP elevation can induce acute ischemia in the ciliary body and iris with severe inflammation in the anterior and posterior chamber, leading to lens zonules injury. Alternatively, zonulopathy could be a consequence of overstressing from the heavy lens or increased posterior pressure (*Zhang et al., 2023*). In summary, occult zonulopathy contributes to the risk of APAC, and lens zonules would be damaged by an IOP elevation attack, creating a vicious circle. A further prospective study would be helpful to illustrate the association between occult zonulopathy and the duration of high IOP.

There are several limitations in this study. First, this study excluded bilateral APAC patients, PACS, and PACG patients. Previous reports indicated that 39.1% of PACG patients and 15.3% of PAC and PACS patients were detected with zonulopathy during cataract surgery (*Zhang et al., 2023*). They may provide insights into the association between zonulopathy and chronic angle closure. Bilateral APAC patients, despite being a minority of all APAC patients, might experience different mechanisms of angle closure from unilateral APAC patients. Second, 21 contralateral eyes of APAC eyes underwent laser peripheral iridectomy (LPI) treatment because of their mild cataract or refusal of surgery. Only six contralateral eyes underwent cataract surgery and accepted assessment of lens zonules during surgery, which might influence the conclusion derived from the inter-eye comparison. Third, the diagnosis of occult zonulopathy was qualitative, depending on the indirect signs intraoperatively, which is subjective. Future studies will benefit from new techniques that allow direct examination of lens zonules for a more quantitative evaluation. Fourth, ocular traumatic history was based on medical history and possible signs of trauma. Minor ocular trauma in children or teenagers might be omitted, although patients with a history or signs of ocular trauma have been excluded. Finally, IOP-lowering treatments might influence ACD because of the changes in IOP level. And the maximum and duration of high IOP were excluded in this study due to their lack of reliability and accuracy, because it was unreliable to obtain IOP in the presence of corneal edema, define maximum IOP in the absence of pre-medical IOP, and record the duration of elevated IOP relying on APAC patients' recall. Further studies are required to validate the findings of this study in other ethnic populations and determine the causality between occult zonulopathy and APAC.

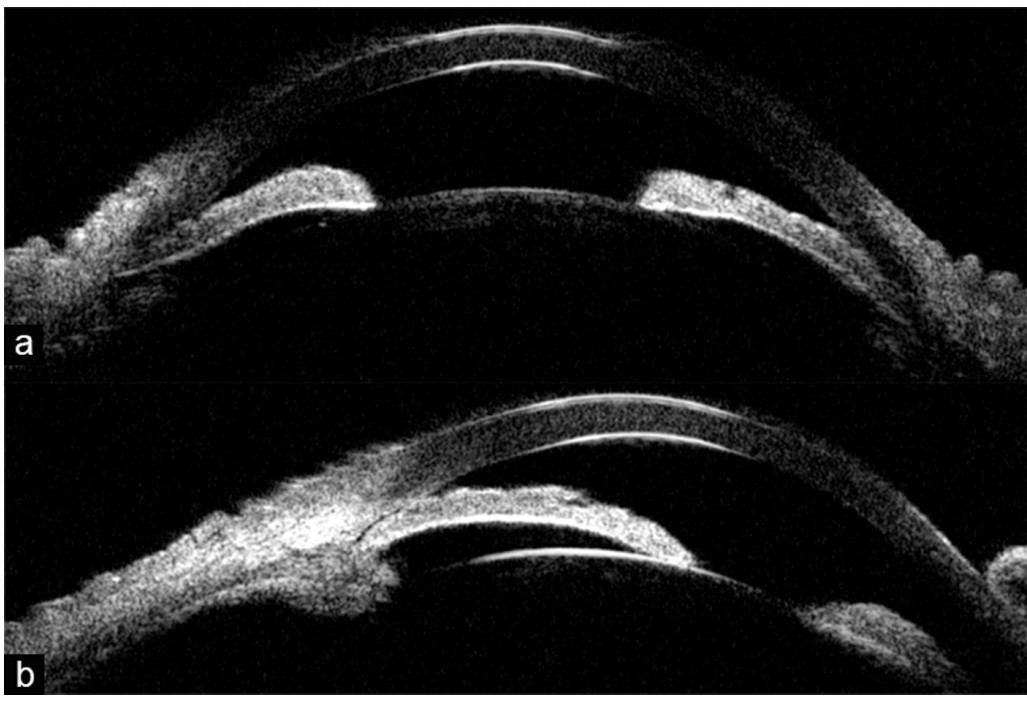

**Figure 3** **(A–B) Pupillary block and anteriorly placed lens showed in ultrasound bio-microscope measurement (UBM).** The iris closely contacts the anterior surface of the lens, rising forward with a great iris curve. A shallow anterior chamber depth is present with a lens closing to the anterior.

## CONCLUSIONS

APAC eyes had a higher incidence of occult zonulopathy than the contralateral and control eyes. Occult zonulopathy was significantly associated with APAC. Occult zonulopathy could be a risk factor for APAC, as it might result in an anterior shifting of the lens.

## ACKNOWLEDGEMENTS

We would like to express our appreciation for the contributions of all the participants in this study.

### Funding

This work was supported by the Hunan Engineering Research Center for Glaucoma with Artificial Intelligence in the Diagnosis and Application of New Materials, China (Grant No.2023TP2225 to Xuanchu Duan), the Natural Science Foundation of Hunan Province, China (Grant No. 2023JJ70014 to Xuanchu Duan), the Science and Technology Foundation of Aier Eye Hospital Group, China (Grant No. AR2206D5 to Xuanchu Duan), the Changsha Municipal Natural Science Foundation, China (Grant No. kq2208495 to Xuanchu Duan), and the Aier Glaucoma Institute. There was no additional external funding received for

this study. The funders had no role in study design, data collection and analysis, decision to publish, or preparation of the manuscript.

## Grant Disclosures

The following grant information was disclosed by the authors:

Engineering Research Center for Glaucoma with Artificial Intelligence in the Diagnosis and Application of New Materials, China: 2023TP2225.

Natural Science Foundation of Hunan Province, China: 2023JJ70014.

Science and Technology Foundation of Aier Eye Hospital Group, China: AR2206D5.

Changsha Municipal Natural Science Foundation, China: kq2208495.

Aier Glaucoma Institute.

## Competing Interests

The authors declare there are no competing interests.

## Author Contributions

- Jiawei Chen conceived and designed the experiments, performed the experiments, analyzed the data, prepared figures and/or tables, authored or reviewed drafts of the article, and approved the final draft.
- Xiang-Ling Yuan performed the experiments, analyzed the data, prepared figures and/or tables, authored or reviewed drafts of the article, and approved the final draft.
- Xinyue Zhang performed the experiments, prepared figures and/or tables, authored or reviewed drafts of the article, and approved the final draft.
- Yanjun Huang performed the experiments, authored or reviewed drafts of the article, and approved the final draft.
- Xiaona Huang performed the experiments, authored or reviewed drafts of the article, and approved the final draft.
- Xuanchu Duan conceived and designed the experiments, performed the experiments, authored or reviewed drafts of the article, and approved the final draft.

## Human Ethics

The following information was supplied relating to ethical approvals (i.e., approving body and any reference numbers):

This study has been approved by the Medical Ethics Committee of the Sixth Affiliated Hospital of South China University of Technology (Foshan Nanhai District People's Hospital; approval number: 2023008).

## Data Availability

The raw data are available in the Supplementary File.

## Supplemental Information

Supplemental information for this article can be found online at http://dx.doi.org/10.7717/peerj.19330#supplemental-information.

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
