# Peer review of "Occult zonulopathy detected during cataract surgery in patients with acute primary angle closure: a retrospective study"

_PeerJ, doi:10.7717/peerj.19330_

## Round 0.1 · original submission · Major Revisions

Reviewers 2 and 3 have both commented on the small sample size and limited number of clinical features, which together have implications on the ability to draw meaningful conclusions

Unless these issues can be resolved (for example by expanding the sample; or providing a more robust statistical justifications; or including additional parameters) it is unlikely that the article can move forward.

·

Basic reporting

The article uses clear, unambiguous, technically correct text. The article conform to professional standards of courtesy and expression.
Literature references, sufficient field background/context are provided.
The article includes sufficient introduction and background to demonstrate how the work fits into the broader field of knowledge. Relevant prior literature are appropriately referenced.
Professional article structure, figures, tables are presented. But Figure 1 needs to amend on the flow chart, numbers are not matching up down the arrow.
The structure of the article conforms to an acceptable format of ‘standard sections’.

Figures are relevant to the content of the article, of sufficient resolution, and appropriately described and labeled.

Experimental design

Original primary research is within Aims and Scope of the journal.
Research question is well defined, relevant & meaningful. It is stated how research fills an identified knowledge gap.

Rigorous investigation performed to a high ethical standard.
The investigation conducted was not to a high technical standard.
The research conducted in conformity with the prevailing ethical standards in the field.
Methods described are with sufficient detail & information to replicate, and with sufficient information to be reproducible by another investigator.

Validity of the findings

Conclusions are well stated, linked to original research question & limited to supporting results.

Additional comments

Figure 1 needs to amend on the flow chart, numbers are not matching down the arrow. The left most column, N=27 goes down to N=264
The 3rd row of Figure 1 needs to be re-arranged.
The investigation conducted was not to a high technical standard. This retrospective case-control without blinding study is of small sample size over the case group.
Authors mentioned that one of the exclusion criteria is history of surgical or laser peripheral, iridectomy, Were patients with argon laser peripheral iridoplasty included? If so, the laser may have contributed to the zonulopathy.
All the ophthalmological examinations were performed under a transparent cornea and IOP </= 30 mmHg. How were APAC cases treated to lower the IOP to </=30? And cataract surgery after the APAC attack may cause the additional wrinkling of anterior lens capsules or floppy capsular bag. Because lens epithelial cells are swollen upon APAC, and glaucomfleckin may form after.
Lam WY, Au SCL. Glaukomflecken: The classic and uncommon ocular sign after acute primary angle closure attack. Vis J Emerg Med. 2023 Apr;31:101702. doi: 10.1016/j.visj.2023.101702. Epub 2023 May 13. PMID: 37206366; PMCID: PMC10182742.

Reviewer 3 ·

Basic reporting

no comment

Experimental design

The authors contribute an interesting manuscript describing the association of occult zonulopathy detected during cataract surgery with acute APAC and investigate the biometric characteristics of APAC patients with or without occult zonulopathy. However, I am concerned that there are some major problems about the experimental design, and we can’t acquire enough information base on the current study.
1. The sample of APAC subjects is limited. The comparison of APAC eyes with occult zonulopathy and without occult zonulopathy in biometric parameters have limited significance. And some valuable parameters may have been missed.
2. As this is a retrospective case-control study, we can’t explore whether occult zonulopathy contributes to the development of APAC or the APAC cause occult zonulopathy. Thus, more clinical features about the APAC should be analyzed, such as the max IOP, duration of high IOP, treatments, etc.
3. Compared to contralateral eyes, shallower central anterior chamber depth, lens position and relative lens position closer to the anterior were found in APAC eyes both with and without occult zonulopathy. This suggested that all of this significant difference might be related to the occurrence of APAC but not occult zonulopathy.

Validity of the findings

no comment

Additional comments

no comment

·

Basic reporting

Please for the authors and editors review the English . Some suggestions have already been made.
Please add one study and consider adding to the comments section

Experimental design

No comment

Validity of the findings

Pleased consider adding comment regarding possible genetic causes that may explain Zonulopathy
Please consider adding comment regarding ethnicity and future studies in different populations.

---

## Round 0.2 · Minor Revisions

Dear authors,

Thank you for your resubmission and efforts. While you have demonstrated the study's validity with statistical content and explanations, i think one of the reviewers appears to want an actual an attempt to address the perceived limitations of the study. While the authors mentioned that the study is not aimed at determining causality, you could further clarify or highlight this in the manuscript. You can explicitly state that the primary goal is to explore associations and propose potential links between occult zonulopathy and APAC, leaving causality for future studies. Also, you could add a brief explanation that and how the study's design was meant to explore potential associations rather than establish definitive cause-and-effect relationships. A mention of why this approach is relevant for understanding the pathophysiology of APAC could help contextualize the study’s aims. You could provide more detailed information on how the sample size was calculated, including the effect size used, the alpha level (0.05), and why this specific sample size was deemed sufficient. This would help reviewers and readers understand the robustness of the analysis better. (I think it was in the rebuttal but not in the manuscript itself). If feasible, you can include a sensitivity analysis showing how changes in sample size or variations in certain parameters (like effect size or assumptions about prevalence) could impact the findings. This could demonstrate that their results remain stable even with variations in sample size or data assumptions.

Moreover, though the authors mention that they couldn’t reliably measure IOP and other factors, they could expand this discussion in the limitations section, explaining how their clinical setting or data collection challenges impacted their ability to measure those parameters. You should also mention any potential confounders considered (e.g., age, underlying conditions, prior treatments) and why these were or were not included in the analysis. This would add depth to their justification and make it clear that they have accounted for other possible variables.

The authors mentioned that occult zonulopathy might cause subtle changes in ocular structures that aren't statistically significant. To further address this concern, they could add more context, perhaps referencing other studies that highlight the challenges of detecting subtle changes in biometric measures or using statistical methods that are sensitive to small effects (e.g., regression models or non-parametric tests) (or better, to add in the supplementary data that show exactly this).

In addition to clarifying the study's aims, the authors could better discuss how even small, non-significant changes in biometric parameters could still be relevant clinically. They could explain how these findings might inform clinical practice, even if the results aren’t statistically significant, particularly for surgeons when assessing APAC patients.

The authors could consider conducting subgroup analyses, if appropriate, to see whether certain characteristics (e.g., severity of zonulopathy, age, or other clinical variables) might reveal patterns that weren't evident in the overall sample. (also, if not already considered, show that you considered different statistical approaches to address potential limitations in the data; if and where relevant).

In order to continue to highlight the attempted overcome of this barrier or limitation, the authors can further still expand their discussion by comparing their findings with similar studies in the literature, especially those that also used biometric measurements in APAC or zonulopathy.

The reviewer raised concerns that the biometric differences found (such as shallower anterior chamber depth and lens position closer to the anterior) might be related to APAC rather than occult zonulopathy. The authors could provide additional references to support why these biometric markers could still be associated with zonulopathy, even if APAC itself is a confounding factor. You can also suggest how future studies might focus on these variables to better distinguish the role of zonulopathy.

So, altogether, i believe that by refining the manuscript for clarity, especially the methods and results sections; clearer explanations of key concepts or aspect and the logic behind the study design could help to mitigate misunderstandings and overcome this standoff. (and showing, although non-significant data results in the supplementary is advisable, in this situation)

·

Basic reporting

The sample size calculation is now listed out. My concerns are addressed.

Experimental design

Reviewers' comments are addressed.

Validity of the findings

Flowcharts are now corrected, my comments are addressed.

Reviewer 3 ·

Basic reporting

This study aimed to determine the association between occult zonulopathy detected during cataract surgery and APAC and to investigate the biometric characteristics of APAC patients with or without occult zonulopathy. However, the association between occult zonulopathy and APAC, as well as the biometric characteristics of APAC patients with occult zonulopathy were both not clearly articulated.

Experimental design

Adjust the experimental purpose and expand the sample size

Validity of the findings

Adjust the experimental purpose and expand the sample size

·

Basic reporting

Thank you for the improvements

Experimental design

Very important contribution to the literature

Validity of the findings

Valid

---

## Round 0.3 · accepted · Accept

Dear authors.
congratulations! It was not easy but i think your manuscript is now ready for publication.

Reviewer 3 ·

Basic reporting

The authors have done an good job of addressing the questions and concerns I raised in my initial review. They have made thoughtful revisions and provided clear explanations for their changes. In my opinion, the manuscript now meets the standards required for publication and is ready to be considered for acceptance.

Experimental design

none

Validity of the findings

none

Additional comments

none